# OpenReview forum: "Martingale Score: An Unsupervised Metric for Bayesian Rationality in LLM Reasoning"
_NeurIPS.cc/2025/Conference — NeurIPS 2025 poster_

### Official Review · Reviewer_K3qf · 2025-06-15

**Clarity:** 2
**Significance:** 3
**Originality:** 3
**Rating:** 2
**Confidence:** 5

**Summary:**

In this paper, the authors study belief entrenchment in LLMs (i.e., current belief predicts future belief updates), proposing a framework to evaluate it in open-ended problems in an unsupervised way. Compared to previous work, they focus more on truth-seeking, where LLMs find and evaluate information. Their approach borrows from a recent tradition of applying Bayesian techniques in the study of LLMs by leveraging the martingale property and score. Their results show positive martingale scores caused by belief entrenchment, which harms accuracy, and that debates mitigate the effect.

**Questions:**

- Why don’t you use API tools? Not a big deal, but it might help make it more robust.
- You should remove the instructions from the checklist, as they say.
- Why does part of the prompt in “The Prompt for Belief Measurement” in Appendix B.2 constantly repeat?
- Line 188: Not a big deal, but we know that some new knowledge can be transferred during post-training.
- Line 225: You seem to be using zero-shot CoT, and I’d cite “Large Language Models are Zero-Shot Reasoners” too
- Giving the results with DeepSeek R1, it’d be great to see results for other reasoning models like o4-mini to test whether that’s a key difference. I’m not asking for more experiments, but this could be mentioned in future work.

**Ethical Concerns:**

["NO or VERY MINOR ethics concerns only"]

**Final Justification:**

I think the idea in this paper is really interesting, and I was willing to increase my initial score (Borderline Reject) if my concerns were addressed. However, **the rebuttal was not convincing at all, so I'm decreasing my score** (Reject). I replied to their rebuttal comments very early on in the discussion period, and I heard back after 5 days. I've considered their last comments, but I believe the evaluation is flawed.

There are weaknesses that I believe can be addressed, such as the writing (lots of typos and lack of clarity), figures (no good explanation of the process or very low quality), and reporting statistical effect sizes.

However, **I'm most concerned about the analysis.** Looking at Figure 4, it's evident that linear regression might not be adequate in this case. There seems to be nonlinearity and/or heteroscedasticity, which make the numerical results irrelevant.

In the rebuttal, the authors reported results controlling for outliers, but first did so strategically only for the cases where the issues I mentioned are not concerning. This seems to have been a misunderstanding; in the end, they reported all the results. However, there's only positive correlation in one setup.

They have changed the outlier removal criteria and included more trajectories because their analysis was underpowered. Moreover, they claim that looking at Debate results and CoT for OpenReview is not very relevant because those datasets are noisy. Coincidentally, there's no positive correlation for those, which means that they're asking the reviewers to disregard the bad results and only consider the good ones.

**I urge all reviewers and AC to review this thoroughly.**

**Limitations:**

Yes, the paper describes limitations in the conclusion.

**Paper Formatting Concerns:**

I have some concerns about the writing and figures that I have included under weaknesses.

**Quality:**

3

**Strengths And Weaknesses:**

**Conditional on addressing all the weaknesses, I’d increase the score.**

I think this work studies a relevant problem and is useful for the ML community, but there are things that need to be improved for me to recommend its acceptance.

## Strengths

- The paper studies an important problem for safely deploying models and agents, and the experiments can be used to test belief entrenchment in realistic applications.
- The authors analyze results for different models (both open and closed source, reasoning models and others), different prompt strategies, and different realistic applications (as previous work has focused on synthetic data or simpler problems).

## Weaknesses

### Writing

The writing in the abstract and introduction could be a lot clearer. For example, in the abstract, “belief entrenchment” is mentioned twice but not defined until line 16. Stylistically, it’d be better to start the Appendix with a section title instead of a figure. Here is a list of typos I found, but there could be more, so please proofread:

- Typo in line 123 “suffers”
- Typo in line 559 “your”
- Typo in line 170 “consists”
- Typo in line 171 with extra space before comma
- Typo in line 172 with “an” instead of “a”
- Line 275 needs fixing
- In Line 237, you need either Appendix B or “in the Appendix”.

### Figures

Most figures in the paper need to be improved. Figures 4, 5, 6, and 7 have tiny fonts, and it’s almost impossible to read them. Figures 4 and 5 could take less space (e.g., Figure 5 can go side by side) to have enough for extra analysis, or other details currently in the Appendix.

Figures 1, 2, and 3A don’t communicate well the problem you’re tackling. For example, Figure 1 is an opportunity to summarize visually your problem, and I think it’s missing the mark. The reasoning steps should be defined as such, and the right side (Weak No, No, Strong No, etc) is not clear where it’s coming from.

The caption of Figure 5 could include more information (e.g., domain details). Similarly, captions in Figures 1 and 2 should elaborate more.

### Analysis

The analysis in Table 1 would benefit from reporting statistical significance at different levels (e.g., typically * (p<0.05), ** (p<0.01), *** (p<0.001)) and their effect size. Without that, it’s hard to say whether statistically the effect of belief entrenchment is that relevant (e.g,. large sample size can lead to small p-values even without significant results).

In Figure 4 and lines 264-266, you claim “We observed that the Martingale Score and Brier score are positive correlated across all setups.” If you look at Figure 4, it seems clear that you’re dealing with a problem of nonlinearity or heteroscedasticity. A few outliers seem to have a lot of leverage on the regression results, as you can clearly see for CoT on OpenReview (red) with that point at (0.15, 0.4). I’d study the leverage of highly influential points and report regression results after that. My impression is that you won’t have meaningful results to claim a positive correlation, except for CoT on Forecasting.

---

> ### Author Rebuttal · Authors · 2025-07-31
>
> Thanks for the review! We highly appreciate the effort and will respond to specific comments.
>
> ### Clarifying the statistical significance and effect size of our findings
>
> Reporting statistical significance at different levels is a very valid constructive point. We will adopt it in the final publication.
>
> We thank the reviewer for this important suggestion. To better demonstrate the practical relevance of our findings, we will report the coefficient of determination (R2) as a standardized measure of effect size.
>
> The Martingale Score (the unstandardized regression coefficient, β1) is itself a direct measure of the effect, but supplementing it with R2 will clarify the proportion of variance in belief updates explained by the prior belief. For instance, in the Gemini 2.0 Flash-NoPrompt-CoT-Forecasting setup, the R2 is 0.0589, indicating the prior belief accounts for approximately 6% of the variance in the update.
>
> Most importantly, we argue that the practical relevance of the Martingale Score is most clearly validated by its **strong correlation with prediction accuracy** (the Brier score), as shown in our Figure 4. This direct link between a higher Martingale Score and a tangible drop in performance confirms its practical importance. We will add the R2 values to our results table and clarify this discussion in the final manuscript.
>
> ### Clarifying the correlation between Martingale Score and Brier score
>
> We thank the reviewer for their sharp observation, as positive correlation does not universally similarly apply to all setups.
>
> Regarding your concern on outliers being overly influential - we conducted three other series of regressions:
>
> - 1. (Light-handed) We remove the 10 most significant outliers (out of 78 points in total) as measured by Cook's distance and re-do all regressions.
> - 2. (Heavy-handed) We remove the 30 most significant outliers as measured by Cook's distance and re-do all regressions.
> - 3. (Conservative): Statistical threshold (4/n Cook's distance)
>
> - The table below summarizes the results, specifically concerning the linear coefficient of the Martingale Score (the dependent variable being Brier score) when under the Debate regime.
> - Ignoring the “+ Reasoning + Interaction” the row given its extremely low statistical power and high p-value, overall, removing outliers does **not** change the sign of the coefficient. This can be confirmed in one case with statistical significance, and another with marginal significance.
>
> *Debate x Martingale Score Interaction Effects Across All Approaches (n=78)*
>
> | Model Specification | No Outlier Handling | Light (≤10) | Heavy (≤30) | Conservative |
> |---------------------|-------------|-------------|-------------|--------------|
> | **+ Reasoning + Interaction** | +0.064 (p=0.861) | +0.409 (p=0.199) | +0.030 (p=0.861) | +0.210 (p=0.565) |
> | **+ Domain + Reasoning + Interaction** | -0.422 (p=0.209) | **-0.547 (p=0.060)** | **-0.472 (p=0.040)** | -0.363 (p=0.239) |
> | **+ Model + Interaction** | -0.481 (p=0.132) | -0.378 (p=0.130) | -0.236 (p=0.179) | -0.252 (p=0.374) |
> | **+ Prompt + Interaction** | -0.193 (p=0.542) | -0.111 (p=0.670) | -0.313 (p=0.133) | -0.195 (p=0.495) |
>
> *Note: Light, heavy, and conservative methods show robust results consistent with no outlier handling.*
>
> The discrepancy among setups on their martingale-brier scores relations
>
> While our results vary across setups, this is expected as the ability to validate the Martingale Score depends on the quality of the experimental conditions.
>
> Our most convincing results come from the setups best suited for this analysis. For example:
>
> - **Chain-of-Thought (CoT) is more trustworthy than Debate** due to a significantly larger sample size.
>
> - **The Forecasting domain is more reliable than OpenReview** because its ground-truth signal is objective and less noisy than the subjective decisions of a few human reviewers. [1]
>
> The primary challenge is finding data with reliable ground truth that the LLM has not seen during training. Forecasting is the ideal domain for this purpose. The strong results obtained in this domain provide the core validation for the martingale score as a meaningful, unsupervised measure of belief entrenchment. In the following research, we hope to find more domains where martingale scores can meaningfully predict either ground truth labeling or collective voted judgments if there is no ground truth. Two domains we plan to run: OpenReview but to assess the relation between martingale score and citation number (as this more closely aligns with collective voted judgments) and Reddict r/AITA (also collective voted judgments) [2]
>
> ### Other suggestions
>
> Also, we very much appreciate that you spend time identifying typos and making suggestions about writing and figures. **All the suggested changes will be reflected in the final manuscript.**
>
> ### Reference
>
> [1] Beygelzimer et al. Has the Machine Learning Review Process Become More Arbitrary as the Field Has Grown? The NeurIPS 2021 Consistency Experiment
>
> [2] Cheng et al. Social Sycophancy: A Broader Understanding of LLM Sycophancy

---

> ### Author Response · Authors · 2025-08-06
> **[1/2] Full Results Beyond Debate**
>
> ### Sample size
>
> Prior to removing outliers, it's 84 in total. 21 in CoT+Forecasting, and likewise for the remaining three setups.
>
> ### Full results
>
> - **Table 1 (CoT, both domains):** **Positive.** With-outlier-removal results and no-outlier-removal results **consistently agree in sign and are positive**. Consistent with Figure 4.
> - **Table 2 (CoT, OpenReview only):** **Weakly positive.** Light and heavy both consistently (100%) positive, while medium leans (75%) negative. No-removal leans positive but without stat significance.
> - **Table 3 (debate, both domains):** **Ambiguous.** Heavy outlier removal is consistently (100%) positive, while medium outlier removal leans (80%) negative. Light outlier removal gives mixed results.
>
> As rows in the tables, we include all regression setups that receive non-trivial $R^2$ in Figure 5. Bold means marginally significant, bold+italic means significant. 'Light' outlier handling removes 3-5 samples according to a $4/n$ Cook's distance threshold; 'medium' always removes 10; and 'heavy' always removes 30.
>
> As a next step, we will look into the cause of ambiguity especially in the debate case.
>
> ---
>
> **Background information:**
>
> - In our final comment (an appendix), we present 5 tables: 1-3 described above, and 4-5 being legacy results that we share for transparency. For reproducibility, we are happy to share the raw data (n=84) upon request in a new comment, but are omitting them here due to excessive length.
> - We produced table 4 containing results for CoT, and table 5 containing CoT+OpenReview. Notably, table 4 presents **much more positive and consistent results** than those in our rebuttals.
>   - Re the question on misleading results: **We hope these positive & consistent results show that we weren't attempting to mislead.** In fact, our decision to only report debate results was a mistake when interpreting the request - we had an internal miscommunication when performing regressions to obtain results for the request here and Reviewer G3H9's questions, the latter specifically asking about debate. The full results in table 1-5 demonstrate our transparency.
> - Outside of table 4, both table 5 and the old one in our rebuttal suffer from an extreme lack of statistical power, and it's hard to reach confident conclusions from them. This is because each data point (i.e. Martingale score for a setup) is computed from only 100 trajectories, so for high-variance domains like OpenReview, and few-step reasoning modes like debate, the Martingale score ended up highly comtaminated with noise.
> - As such, over the past few days, we performed a full re-computation of our data **with 6x OpenReview trajectories-per-sample and 7.5x debate trajectories-per-sample** (i.e. we independently sample more reasoning trajectories for each question), and then carried out regressions. Now each of the debate or OpenReview samples in regression is backed by, on avergage, 6x or 7.5x reasoning trajectories.
>     - Table 1-3 are results of regressions performed on this dataset with strengthened statistical power.

---

> ### Author Response · Authors · 2025-08-06
> **[2/2] Generality of the Martingale Score**
>
> > I still have issues with Figure 4 that haven't been addressed. You claim positive correlations between Martingale and Brier scores, but I think you might only be able to do so for one particular case. Yet, as you said, "the practical relevance of the Martingale Score is most clearly validated by its strong correlation with prediction accuracy (the Brier score)".
>
> We'd like to note that we see weakly positive results in CoT+OpenReview as well, in addition to the strongly positive results in CoT+Forecasting. We believe the weaker results for OpenReview can be explained by issues with the OpenReview task itself - more on this below.
>
> We would like to first clarify some issues with Debate and OpenReview, and why the presentation of Figure 4 may cause confusion; then we hope to address the concern with regard to the generality of the Martingale Score.
>
> ### Issues with Debate and OpenReview
> - We believe CoT is a much better testbed than Debate is, and we should focus most of our attention on CoT. Martingale is to evaluate the belief entrenchment of a given reasoner, whereas in debate two debaters intentionally present opposite views. Even if the two debaters are entrenched by their own parametric beliefs, there are always views from both sides present to the judges. Naturally, belief entrenchment, even if exists, is hardly observable in such setups, and thus Martingale scores for debate tend to be dominated by stochastic noise.
> - OpenReview has very noisy ground-truth signals (i.e. acceptance decisions). The NeurIPS 2021 review consistency experiment [4] demonstrates that there is shockingly large randomness in the acceptance decisions, e.g. spotlight papers have >50% chance of being rejected if independently reviewed again. Hence OpenReview may not be a great testbed for Martingale Score either.
> - There are 4 plots in Figure 4. Admittedly, only Forecasting + CoT setups have achived results that strongly support that claim "the practical relevance of the Martingale Score is most clearly validated by its strong correlation with prediction accuracy (the Brier score)". However, we'd like to note that **forecasting is NOT merely a dataset within a task, but an ML task in itself, covering a diverse range of real-world subjects.**
>
> ### The Versatility and Unique Position of Forecasting as a Task
> - Using LMs for event forecasting is, in itself, a machine learning task that has seen a long line of research papers ([1,2,3] to name a few) and tightly connected with the general world-modeling ability of LLMs.
> - As an application of the Martingale score, we explored a new path to evaluating performance on such a task: testing inference-time cognitive bias by incentivise sound forecasting reasoning, instead of incentivising memorization of related facts. Forecasting is unique because no related facts to be memorized if LLM training cut-off dates were earlier than the dates when forecasting questions are resolved.
>
> ### Martingale Score is Generalizable
>
> - We still hold the belief that Martingale Score is generalizable. Our Forecasting dataset includes a diverse range of domains including economy/business, environment/climate, elections/geopolitics, technology/AI, and more. LM-based forecasting is one general methodology to deal with many problems in different problem domains.
> - Forecasting tasks are about incorporating evidence in light of uncertainties. Such a capability is central to general reasoning, and it is precisely what Martingale Score measures.
>
> ---
>
> [1] Advancing Event Forecasting through Massive Training of Large Language Models: Challenges, Solutions, and Broader Impacts
>
> [2] Approaching Human-Level Forecasting with Language Models
>
> [3] Evaluating LLMs on Real-World Forecasting Against Expert Forecasters
>
> [4] Has the Machine Learning Review Process Become More Arbitrary as the Field Has Grown? The NeurIPS 2021 Consistency Experiment

---

> ### Author Response · Authors · 2025-08-06
> **Appendix A: Tables for Regression Results**
>
> **Table 1: Full Martingale-Brier Causal Relations for ChainOfThought (OpenReview + Forecasting)**
>
> | Reg. Variables | No Outlier Handling | Light | Medium | Heavy |
> |-------|---------------------|-------|--------|-------|
> | Basic  | **0.632 (p=0.08,n=84)** | 0.509 (p=0.16,n=81) | -0.105 (p=0.74,n=74) | **___-0.354 (p=0.03,n=54)___** |
> | + Domain  | **___0.979 (p<0.01,n=84)___** | **___0.948 (p<0.01,n=81)___** | **___0.738 (p<0.01,n=74)___** | -0.110 (p=0.59,n=54) |
> | + Domain + Model  | **___1.119 (p<0.01,n=84)___** | **___1.032 (p<0.01,n=81)___** | **___0.674 (p<0.01,n=74)___** | -0.079 (p=0.71,n=54) |
> | + Domain + Prompt  | 0.450 (p=0.22,n=84) | 0.518 (p=0.16,n=81) | **0.696 (p=0.05,n=74)** | -0.034 (p=0.90,n=54) |
> | + Domain + Model + Prompt  | 0.608 (p=0.12,n=84) | 0.635 (p=0.10,n=81) | **0.604 (p=0.08,n=74)** | 0.005 (p=0.99,n=54) |
>
> ---
>
> **Table 2: Full Martingale-Brier Causal Relations for ChainOfThought (OpenReview only)**
> | Reg. Variables | No Outlier Handling | Light | Medium | Heavy |
> |-------|---------------------|-------|--------|-------|
> | Basic  | 0.120 (p=0.81,n=42) | **0.604 (p=0.06,n=37)** | -0.282 (p=0.51,n=32) | **___0.109 (p<0.01,n=12)___** |
> | + Model  | 0.562 (p=0.35,n=42) | **___0.700 (p=0.03,n=37)___** | 0.314 (p=0.47,n=32) | 0.068 (p=0.11,n=12) |
> | + Prompt  | -0.036 (p=0.95,n=42) | 0.389 (p=0.35,n=37) | **___-1.753 (p<0.01,n=32)___** | **0.140 (p=0.06,n=12)** |
> | + Model + Prompt  | 0.332 (p=0.63,n=42) | 0.546 (p=0.26,n=37) | **___-1.649 (p=0.01,n=32)___** | Over-parameterized (n=12) |
>
> ---
>
> **Table 3: Full Martingale-Brier Causal Relations for Debate (OpenReview + Forecasting)**
> | Reg. Variables | No Outlier Handling | Light | Medium | Heavy |
> |-------|---------------------|-------|--------|-------|
> | Basic  | 0.290 (p=0.11,n=84) | **0.290 (p=0.07,n=81)** | **___0.290 (p=0.02,n=74)___** | **___0.267 (p<0.01,n=54)___** |
> | + Domain  | -0.171 (p=0.38,n=84) | -0.227 (p=0.18,n=81) | **___-0.242 (p=0.04,n=74)___** | **0.149 (p=0.07,n=54)** |
> | + Domain + Model  | -0.320 (p=0.13,n=84) | **___-0.403 (p=0.02,n=81)___** | **___-0.370 (p<0.01,n=74)___** | 0.067 (p=0.47,n=54) |
> | + Domain + Prompt  | 0.157 (p=0.49,n=84) | 0.053 (p=0.80,n=81) | -0.127 (p=0.43,n=74) | 0.195 (p=0.12,n=54) |
> | + Domain + Model + Prompt  | 0.049 (p=0.85,n=84) | -0.135 (p=0.57,n=81) | **-0.295 (p=0.08,n=74)** | 0.078 (p=0.60,n=54) |
>
>
> ---
>
>
> **Table 4: Legacy (Underpowered) Martingale-Brier Causal Relations for ChainOfThought (OpenReview + Forecasting)**
> | Reg. Variables | No Outlier Handling (n=78) | Light (≤10) (n=68) | Heavy (≤30) (n=48) | Conservative (n=72-75) |
> |---------------------|-------------|-------------|-------------|--------------|
> | **+ Reasoning** | -0.181 (p=0.579) | -0.527 (p=0.079) | -0.130 (p=0.425) | -0.328 (p=0.331) |
> | **+ Domain + Reasoning** | **+0.865 (p=0.019)** | **+0.999 (p=0.003)** | **+0.860 (p=0.004)** | **+0.918 (p=0.008)** |
> | **+ Model** | **+0.822 (p=0.021)** | **+0.719 (p=0.013)** | **+0.504 (p=0.027)** | **+0.729 (p=0.026)** |
> | **+ Prompt** | +0.592 (p=0.088) | **+0.641 (p=0.032)** | **+0.704 (p=0.003)** | **+0.727 (p=0.024)** |
>
> ---
>
> **Table 5: Legacy (Underpowered) Martingale-Brier Causal Relations for ChainOfThought (OpenReview only)**
> | Reg. Variables | No Outlier (n=36) | Light ≤5 (n=31) | Heavy ≤15 (n=21) | Conservative (n=33) |
> |---------------------|-------------|-------------|-------------|--------------|
> | **+ Reasoning** | +0.152 (p=0.621) | +0.152 (p=0.424) | +0.153 (p=0.215) | +0.152 (p=0.544) |
> | **+ Domain + Reasoning** | +0.152 (p=0.621) | +0.152 (p=0.424) | +0.153 (p=0.215) | +0.152 (p=0.544) |
> | **+ Model** | -0.354 (p=0.251) | -0.279 (p=0.087) | **-0.562 (p=0.006)** | -0.231 (p=0.144) |
> | **+ Prompt** | +0.149 (p=0.603) | +0.236 (p=0.225) | +0.178 (p=0.645) | +0.149 (p=0.510) |

---

> > ### Comment · Reviewer_K3qf · 2025-08-06
> >
> > Thank you for providing all these details. Before diving deeply, I'd like to get more clarity about how you're handling outliers.
> >
> > > Prior to removing outliers, it's 84 in total. 21 in CoT+Forecasting, and likewise for the remaining three setups.
> >
> > > Regarding your concern on outliers being overly influential - we conducted three other series of regressions:
> >
> > > * (Light-handed) We remove the 10 most significant outliers (out of 78 points in total) as measured by Cook's distance and re-do all regressions.
> > > * (Heavy-handed) We remove the 30 most significant outliers as measured by Cook's distance and re-do all regressions.
> > > * (Conservative): Statistical threshold (4/n Cook's distance)
> >
> > Considering this, I'm a bit confused. **Are you detecting outliers based on all 84 points?** I think that would be a mistake, and you need to do it based on each setup separately. If you're already doing it this way, then how did you decide how many outliers to remove? Removing 10 points would be almost 50%, which is definitely not light, and removing 30 would be over 100% of the total points.

---

> > > ### Author Response · Authors · 2025-08-06
> > >
> > > Thank you for the question, and for your efforts in the deep dive.
> > >
> > > - For Table 1 (CoT, both domains) and Table 3 (debate, both domains), we detect outliers based on all 84 points.
> > > - For Table 2 (CoT, OpenReview only), we detect outliers based on the 42 OpenReview points.
> > >
> > > We believe this strategy is sound, since we use Cook's distance as the detection metric, and Cook's distance measures excessive impact on regression results **given a regression formula**. In other words, **for each row of each table, we find the outliers specific to that regression model**.
> > >
> > > Since all of these regression models have the domain covariate & the reasoning mode covariate, it means that we are already finding the outliers for each of the 2x2 setups. We won't consider a sample an outlier just because it is far away from samples *of a different setup*.
> > >
> > > Finally, to clarify our regression method: we use all 84 sample for all regressions in Table 1 & 3, but since we have domain & reasoning mode covariates, both as intercept terms and as slope terms (i.e. interaction terms with the Martingale score variable), the regression process is mathematically equivalent to separate regressions in separate setups.
> > >
> > > Please let us know if you have follow-up questions!

---

> > > > ### Comment · Reviewer_K3qf · 2025-08-07
> > > >
> > > > Thank you, I have one more request before continuing to review the latest results.
> > > >
> > > > How many points remain per setup after handling the outliers (i.e., in light, heavy, and conservative cases)? If you're removing from the total of 84, I imagine there's no guarantee that the outliers are sampled uniformly per setup.

---

> > > > > ### Author Response · Authors · 2025-08-07
> > > > >
> > > > > Thank you for the question!
> > > > >
> > > > > Overall, there isn't significant imbalance between domains (forecasting vs openreview), though there is between reasoning modes (CoT vs debate). Namely, many more outliers tend to be detected in CoT than in debate.
> > > > >
> > > > > Take as example the basic regression model in Table 1 & 3.
> > > > > - Under light, among the 3 outliers removed, 1 is in Forecasting + CoT, 2 are in OpenReview + CoT.
> > > > > - Under medium, among the 10 outliers removed, 4 is in Forecasting + CoT, 6 are in OpenReview + CoT.
> > > > > - Under heavy, among the 30 outliers removed, 9 is in Forecasting + CoT, 15 are in OpenReview + CoT, and 6 are in Debate.
> > > > >
> > > > > This is aligned with impressions from reading the scatter plots: Debate data points tend to more closely follow a Gaussian pattern where outliers tend to be few and insignificant; CoT data in both domains follow more structured, multi-centered distributions, and so outliers are much more common.
> > > > >
> > > > > We therefore believe this differential treatment of setups based on the prevalence of outliers is more appropriate than enforcing equal counts in different setups, as the latter would lead to vastly different distance thresholds for different setups even under the same outlier handling level.

---

> ### Comment · Reviewer_K3qf · 2025-08-07
> **[1/2]**
>
> First of all, thank you for such detailed responses. I hope we have enough time before the deadline to discuss everything. I'm going to summarize the discussion so far for the other reviewers and AC.
>
> **Summary up until now:** My main concern during the initial review was the evaluation, especially issues of nonlinearity and heteroskedasticity in Figure 4. The authors in their rebuttal provided results handling outliers in three conditions (1) light (up to 10 outliers removed), (2) heavy (up to 30 outliers removed), and (3) conservative. However, they did not report results for all setups due to a misunderstanding on their side. Recently, they provided a lot more results with different conditions (1) light (up to 3 outliers removed), (2) medium (up to 10 outliers removed), and (3) heavy (up to 30 outliers removed). Besides changing the criteria for outlier removal, they also found that they were underpowered and included more trajectories in the analysis (with different criteria per setup). The outliers are computed from the total number of points (n=84), which does not guarantee balanced removal across setups (each setup has n=21).
>
> **One of the issues I see with removing outliers based on all points, is the imbalance it creates**. For example, **no outliers are removed for Debate on Forecasting or OpenReview**, except for the few in heavy removal. However, looking at Figure 4, I can see that Debate would benefit from removing some outliers.
>
> > Under heavy, among the 30 outliers removed, 9 is in Forecasting + CoT, 15 are in OpenReview + CoT, and 6 are in Debate.
>
> What do you mean by "6 are in Debate"? Does that include both OpenReview and Forecasting?
>
> The imbalance here becomes a problem, and heavy might be too heavy to even consider these results very useful. For Forecasting + CoT, you're removing almost 50% of points, and for OpenReview + CoT you're removing over 70%. At that point, the regression is meaningless.
>
> What is the justification for changing the outlier removal criteria? That is, going from light, heavy, and conservative to light, medium, and heavy?
>
> > Table 1
>
> The Basic modeling in Table 1 (CoT for OpenReview and Forecasting) shows positive correlation (0.632) when no outliers are removed. It decreases a bit for light (0.509, i.e., removing 3 points), and flips the sign for medium (-0.105, i.e. removing 10 points) and heavy (-0.354, i.e. removing 24 points).
>
> By default, it's not very promising, so it's great that the authors controlled for other variables. When controlling for domain, the consensus is better (except for heavy, but I'd ignore that). **My takeaway is that one of these two setups does not have a positive correlation,** which you can see below as I discuss Table 2.
>
> > Table 2
>
> The Basic modeling (CoT + OpenReview only) goes from 0.120 (no outliers) to 0.604 (light, removes 2 outliers) to -0.282 (medium, removes 6 outliers) to 0.109 (heavy, removes 15 outliers). **My takeaway is that there is no meaningful correlation.** My main reason is that the medium outlier removal is the one that makes the most sense and it flips signs.
>
> > Table 3
>
> The Basic modeling (Debate for both OpenReview and Forecasting) shows no difference for light and medium outlier removal. **The reason behind it is the imbalance in the outlier detection, since no outliers are being removed in this case**. There are some slight changes in heavy outlier removal, but only 6 points were removed.
>
> However, when controlling for domain and other variables, correlations flip. **My takeaway is that these correlations are not meaningful.**
>
> ---
>
> Let me address some of your comments on your results:
>
> > Table 1 (CoT, both domains): Positive. With-outlier-removal results and no-outlier-removal results consistently agree in sign and are positive. Consistent with Figure 4.
>
> I think this is only true for CoT on Forecasting. Please see what I said above under Table 1.
>
> > Table 2 (CoT, OpenReview only): Weakly positive. Light and heavy both consistently (100%) positive, while medium leans (75%) negative. No-removal leans positive but without stat significance.
>
> I think this is not true. Already, no-removal is not significant, heavy removes too many points (over 70%) to be a useful signal; and medium flips.
>
> > Table 3 (debate, both domains): Ambiguous. Heavy outlier removal is consistently (100%) positive, while medium outlier removal leans (80%) negative. Light outlier removal gives mixed results.
>
> To me, this is not ambiguous at all. The outliers are not removing anything (or almost anything), so is expected for Basic to not change at all. However, when controling for other variables correlations reliably flip.

---

> ### Comment · Reviewer_K3qf · 2025-08-07
> **[2/2]**
>
> > Outside of table 4, both table 5 and the old one in our rebuttal suffer from an extreme lack of statistical power
>
> You should've identified beforehand that you were underpowered. It's fine, we all miss things, but now the issue is that including more trajectories increases power, but also can be seen as p-hacking.
>
> > with 6x OpenReview trajectories-per-sample and 7.5x debate trajectories-per-sample
>
> What is the rationale behind 6x vs 7.5x?
>
> > We'd like to note that we see weakly positive results in CoT+OpenReview as well,
>
> Again, I disagree. See my comments above under Table 2.
>
> > in addition to the strongly positive results in CoT+Forecasting
>
> This I believe is correct because of controlling for Domain. Still, would you mind sharing this by itself? Table 1 includes both Forecasting and OpenReview.
>
> > We believe CoT is a much better testbed than Debate is, and we should focus most of our attention on CoT.
>
> I might agree with this, but then the question is, why do you include Debate at all? If we have to ignore it, then you are only evaluating CoT (quite minimal evaluation) and could also be seen as cherry picking. **To me this is a sign of a poor evaluation design.**
>
> > OpenReview has very noisy ground-truth signals (i.e. acceptance decisions). [...] Hence OpenReview may not be a great testbed for Martingale Score either.
>
> **Again, this shows me that there is a flaw in your evaluation**. Otherwise, this just leaves us with only your results for Forecasting and CoT, where you have the largest positive correlation.
>
> > There are 4 plots in Figure 4. Admittedly, only Forecasting + CoT setups have achived results that strongly support that claim "the practical relevance of the Martingale Score is most clearly validated by its strong correlation with prediction accuracy (the Brier score)". However, we'd like to note that forecasting is NOT merely a dataset within a task, but an ML task in itself, covering a diverse range of real-world subjects.
>
> I'm sorry, but this sounds like an ad-hoc justification. If that were true, then why would you need to include all other setups?
>
> > We still hold the belief that Martingale Score is generalizable.
>
> I might agree with this, and I really like the core idea of the paper. However, your evaluation is flawed, and it's not answering this question convincingly.
>
> ---
>
> In your rebuttal, you addressed my most urgent concerns, but missed something that I'd like to know:
>
> > Why does part of the prompt in “The Prompt for Belief Measurement” in Appendix B.2 constantly repeat?

---

> ### Author Response · Authors · 2025-08-09
> **Thank you for the comments**
>
> We'd like to thank the reviewer again for the thoughtful feedback!
>
> > Besides changing the criteria for outlier removal
>
> Quick clarification: we did not change the criteria. The new modes light/medium/heavy are simply a rename of the old conservative/light/heavy modes respectively; you can verify this by checking the definition of light/medium/heavy in our response. We apologize for the confusion!
>
> > One of the issues I see with removing outliers based on all points, is the imbalance it creates.
>
> We agree with this concern. Please see the new Table 1-4 for per-quadrant results, with the imbalance issue fixed by restricting samples to only those in the quadrant. We also adjust medium and heavy modes to percentage-based (15% and 30% respectively) rather than fixed-count, to avoid removing too many samples, while aligning with the original removal counts on the full dataset.
>
> Light mode remains based on a $4/n$ Cook's distance threshold.
>
> > What do you mean by "6 are in Debate"? Does that include both OpenReview and Forecasting?
>
> Yes. And upon your request (which we agree with), we present regression results for each of the 2x2 quadrants in the new Table 1-4.
>
> > Table 1 / Table 2 / Table 3
>
> We agree with you that much of the inconsistency is due to (1) mixing Forecasting and OpenReview in the same table, and (2) outlier imbalance causing unreasonably many removals.
>
> We have tried to address these issues in the new Table 1-4. The verdicts there seem much clearer. Notably, **within each quadrant, significant results now always come with the same sign.**
> - **Forecasting-Only ChainOfThought (Table 1): Positive.**
> - **OpenReview-Only ChainOfThought (Table 2): Leans positive; only partially/marginally significant.** (note the multiple-testing issue here; that said, even the signs of insignificant results are consistent)
> - **Forecasting-Only SelfDebate (Table 3): Leans negative; only partially significant.** (note the multiple-testing issue here)
> - **OpenReview-Only SelfDebate (Table 4): Positive.**
>
> > including more trajectories increases power, but also can be seen as p-hacking.
>
> We acknowledge that it's a valid concern. That said, testing 2 times results in a Bonferroni correction factor of 2 (a conservative estimate given the non-independence between tests; the actual factor is somewhere between 1 and 2). Given that most of our $p<0.05$ results indeed satisfy $p<0.025$, such a correction isn't fatal to our conclusions. We will add such a note to our results in the body of the paper.
>
> > What is the rationale behind 6x vs 7.5x?
>
> Upon examination of the legacy data, we heuristically chose to multiply all OpenReview sample counts by 2x, and all Debate sample counts by 5x (the latter to compensate for the, on average, 4-5x fewer reasoning steps that debate trajectories have). The two factors multiply, so the OpenReview + Debate setup receive 10x.
>
> When summed across OpenReview setups, total sample size becomes (2+10)/2=6x that the original sample size, and for Debate it's (5+10)/2=7.5x. These are "on average" statistics, as we mentioned in our previous response.
>
> In our new Table 1-2, we have leveled the gap between CoT and Debate by multiplying the former by 5x too. Now all Forecasting data points are backed by 1000 trajectories each, and all Debate data points are backed by 2000 trajectories each.
>
> > This I believe is correct because of controlling for Domain. Still, would you mind sharing this by itself? Table 1 includes both Forecasting and OpenReview.
>
> Sure. Please see the new Table 1 attached.
>
> > I might agree with this, but then the question is, why do you include Debate at all?
>
> We started out hoping to demonstrate the Martingale score's cross-task, cross-reasoning mode applicability by having this 2x2 quadrant.
>
> We only realized the issue of noise and statistical power in Debate and OpenReview later on in our analysis, and by that point, removing those results would actually be cherry-picking. For transparency, we decided to include them in the body of our paper too.
>
> > Why does part of the prompt in “The Prompt for Belief Measurement” in Appendix B.2 constantly repeat?
>
> Apologies - there was a formatting error when we created the code block. As you can see, it contains multiple string variables:
> - `belief_eval_judge_prompt`: The prompt itself; the first string in the block; variable name blocked out by accident.
> - `belief_eval_judge_prompt_with_traj`: The prompt itself, but with the per-trajectory evaluation variation. Namely, the judge model reads an entire reasoning trajectory at a time, and assigns belief scores to each step all at once. This reduces within-trajectory random noise in judge assessments. **This is the default prompt design we use in our results.**
> - `additional_info_interlude` / `additional_info_item` / `additional_info_ending`: These are add-on items to the prompt, where you pass things like background information related to the topic. We do not use these by default.

---

> ### Author Response · Authors · 2025-08-09
> **Per-Quadrant Regression Tables**
>
> To level the sample size gap between CoT and Debate, we have multiplied the former's sample size by 5x, making the two equal in statistical power. This changes the numbers in Table 1-2.
>
> For transparency, we include pre-expansion results too in Table 1a-2a at the end. We acknowledge the double-testing concern associated with expanding the sample size, which we hope to mitigate with transparency on the previous results.
>
> **Table 1: Forecasting-Only ChainOfThought (post-expansion: consistent with pre-expansion)**
>
> | Model     |  No Outlier Handling   |  Light      |  Medium     |  Heavy     |
> |-------|---------------------|-------|--------|-------|
> | Basic     |  **___2.899 (p<0.01,n=21)___**  |  **___2.899 (p<0.01,n=21)___**  |  **___2.924 (p<0.01,n=18)___**  |  **___2.991 (p<0.01,n=15)___** |
> | Basic + Model      |  **___3.349 (p<0.01,n=21)___**  |  **___3.538 (p<0.01,n=19)___**  |  **___3.617 (p<0.01,n=18)___**  |  **___3.308 (p<0.01,n=15)___** |
> | Basic + Prompt     |  1.061 (p=0.13,n=21)  |  1.061 (p=0.13,n=20)  |  1.122 (p=0.11,n=18)  |  1.286 (p=0.19,n=15) |
> | Basic + Model + Prompt      |  1.651 (p=0.23,n=21)  |  **2.457 (p=0.06,n=17)**  |  **2.457 (p=0.06,n=18)**  |  **3.490 (p=0.06,n=15)** |
>
> **Table 2: OpenReview-Only ChainOfThought (post-expansion: mixed coef signs pre-expansion -> consistent coef signs post-expansion)**
>
> | Model     |  No Outlier Handling   |  Light      |  Medium     |  Heavy     |
> |-------|---------------------|-------|--------|-------|
> | Basic     |  0.728 (p=0.26,n=21)  |  1.047 (p=0.22,n=18)  |  1.047 (p=0.22,n=18)  |  0.657 (p=0.23,n=15) |
> | Basic + Model      |  1.333 (p=0.12,n=21)  |  **1.820 (p=0.08,n=18)**  |  **1.820 (p=0.08,n=18)**  |  -1.263 (p=0.19,n=15) |
> | Basic + Prompt     |  0.237 (p=0.63,n=21)  |  0.454 (p=0.21,n=19)  |  0.475 (p=0.25,n=18)  |  0.600 (p=0.14,n=15) |
> | Basic + Model + Prompt      |  0.760 (p=0.27,n=21)  |  1.521 (p=0.59,n=15)  |  1.057 (p=0.17,n=18)  |  1.521 (p=0.59,n=15) |
>
> **Table 3: Forecasting-Only SelfDebate**
>
> | Model     |  No Outlier Handling   |  Light      |  Medium     |  Heavy     |
> |-------|---------------------|-------|--------|-------|
> | Basic     |  -0.041 (p=0.22,n=21)  |  -0.033 (p=0.49,n=19)  |  -0.007 (p=0.87,n=18)  |  -0.025 (p=0.52,n=15) |
> | Basic + Model      |  **___-0.082 (p=0.03,n=21)___**  |  **___-0.126 (p<0.01,n=17)___**  |  **___-0.141 (p<0.01,n=18)___**  |  **___-0.107 (p=0.02,n=15)___** |
> | Basic + Prompt     |  0.004 (p=0.90,n=21)  |  0.009 (p=0.79,n=19)  |  0.007 (p=0.83,n=18)  |  -0.000 (p=0.99,n=15) |
> | Basic + Model + Prompt      |  0.008 (p=0.89,n=21)  |  -0.110 (p=0.54,n=15)  |  -0.109 (p=0.36,n=18)  |  -0.110 (p=0.54,n=15) |
>
> **Table 4: OpenReview-Only SelfDebate**
>
>
> | Model     |  No Outlier Handling   |  Light      |  Medium     |  Heavy     |
> |-------|---------------------|-------|--------|-------|
> | Basic     |  **___0.294 (p<0.01,n=21)___**  |  **___0.294 (p<0.01,n=21)___**  |  **___0.193 (p=0.02,n=18)___**  |  **___0.192 (p=0.02,n=15)___** |
> | Basic + Model      |  **___0.229 (p<0.01,n=21)___**  |  **___0.184 (p<0.01,n=18)___**  |  **___0.184 (p<0.01,n=18)___**  |  **___0.200 (p<0.01,n=15)___** |
> | Basic + Prompt     |  **___0.290 (p<0.01,n=21)___**  |  **___0.299 (p<0.01,n=20)___**  |  **___0.328 (p<0.01,n=18)___**  |  **___0.388 (p<0.01,n=15)___** |
> | Basic + Model + Prompt      |  0.040 (p=0.81,n=21)  |  Over-parameterized (n=13)  |  0.184 (p=0.21,n=18)  |  **___0.293 (p=0.03,n=15)___** |
>
> ---
>
> **Table 1a: Forecasting-Only ChainOfThought (pre-expansion)**
>
> | Model     |  No Outlier Handling   |  Light      |  Medium     |  Heavy     |
> |-------|---------------------|-------|--------|-------|
> | Basic     |  **___2.824 (p<0.01,n=21)___**  |  **___2.662 (p<0.01,n=20)___**  |  **___2.889 (p<0.01,n=18)___**  |  **___2.990 (p<0.01,n=15)___** |
> | Basic + Model      |  **___3.218 (p<0.01,n=21)___**  |  **___2.969 (p<0.01,n=20)___**  |  **___2.955 (p<0.01,n=18)___**  |  **___2.849 (p<0.01,n=15)___** |
> | Basic + Prompt     |  0.732 (p=0.37,n=21)  |  0.713 (p=0.40,n=17)  |  0.686 (p=0.41,n=18)  |  0.537 (p=0.54,n=15) |
> | Basic + Model + Prompt      |  2.716 (p=0.18,n=21)  |  3.400 (p=0.15,n=15)  |  2.958 (p=0.25,n=18)  |  3.400 (p=0.15,n=15) |
>
> **Table 2a: OpenReview-Only ChainOfThought (pre-expansion)**
>
> | Model     |  No Outlier Handling   |  Light      |  Medium     |  Heavy     |
> |-------|---------------------|-------|--------|-------|
> | Basic     |  0.120 (p=0.87,n=21)  |  -0.018 (p=0.98,n=19)  |  -0.568 (p=0.37,n=18)  |  0.478 (p=0.36,n=15) |
> | Basic + Model      |  0.705 (p=0.48,n=21)  |  **___-1.700 (p=0.01,n=17)___**  |  **1.523 (p=0.10,n=18)**  |  **___-2.202 (p<0.01,n=15)___** |
> | Basic + Prompt     |  -0.075 (p=0.88,n=21)  |  0.328 (p=0.47,n=18)  |  0.328 (p=0.47,n=18)  |  **0.686 (p=0.09,n=15)** |
> | Basic + Model + Prompt      |  0.657 (p=0.41,n=21)  |  0.865 (p=0.71,n=15)  |  1.108 (p=0.22,n=18)  |  0.865 (p=0.71,n=15) |

---

### Official Review · Reviewer_NJVu · 2025-06-20

**Clarity:** 3
**Significance:** 3
**Originality:** 3
**Rating:** 5
**Confidence:** 3

**Summary:**

The authors introduce an unsupervised, regression-based metric, Martingale Score, designed to measure belief entrenchment in LLMs. The paper argues that despite advancements in reasoning techniques, LLMs often exhibit "belief entrenchment" and "confirmation bias," where their iterative reasoning reinforces existing beliefs rather than promoting their truth-seeking behavior. The proposed Martingale Score quantifies deviations from the Bayesian Martingale property, which dictates that belief updates should be unpredictable from prior beliefs, thus signaling irrational entrenchment. The authors find that such violations are widespread across various LLM models, reasoning techniques (like Chain-of-Thought and Debate), and problem domains. The paper concludes by identifying factors that alleviate belief entrenchment, highlighting the prevalence and impact of this cognitive bias in LLM reasoning. Overall, the paper gives a very fresh perspective. Please refer to the strengths and weaknesses section for some open questions.

**Questions:**

Please refer to the Weakness section for more details.

**Ethical Concerns:**

["NO or VERY MINOR ethics concerns only"]

**Final Justification:**

The authors propose an unsupervised, regression-based metric to measure belief entrenchment in LLMs, which can be used to capture the model’s ability to form new beliefs (different than the parameter beliefs that the model has learned) during inference time through extended reasoning.

**Limitations:**

Please refer to the Weakness section for more details.

**Quality:**

3

**Strengths And Weaknesses:**

**Strengths**

1. The paper presents some very interesting results, including the relation between belief entrenchment and accuracy loss, where the authors show that belief entrenchment consistently predicts accuracy loss and a proxy for reasoning quality.

2. Figure 3 is very nice, where the authors have summarized their entire experimental setup.

3. The authors perform an extensive empirical analysis across diverse families of reasoning and non-reasoning models, highlighting the effectiveness and significance of Martingale scores along with t-test results.

**Weakness and Open Questions**

1. In Line 36, the authors describe that current reasoning improvement techniques aim to help the language models in truth-seeking. This could be argued in different ways. Are the authors using "truth-seeking" as a proxy for correctness? Most mathematical, logical, and commonsense reasoning problems are grounded in training the model to solve the problems correctly.

2. In Line 109, the authors put forward a very strong argument: "If reasoning is non-Bayesian, then reasoning techniques do not help humans seek truth." It seems the authors are again focusing more on the Agentic capabilities of LLMs, where we use their reasoning skills to search for different evidence before addressing a query/task.

3. In Line 130, the authors mention "effective reasoning requires the capacity to update beliefs in response to new evidence" -- doesn't the feedback and training, and the underlying principle of the autoregressive nature of LLMs ensure this by default?

4. In Section 5.2, how well is the LLM judge calibrated to score revealed beliefs?

5. In Section 5.3, the authors target events that were resolved after the model's knowledge cut-off. Isn't this exactly something that the model cannot answer from prior knowledge? So, this is against belief entrenchment?

6. Minor: Figure 2 is unclear and doesn't help in understanding the main takeaways.

---

> ### Author Rebuttal · Authors · 2025-07-31
>
> Thank you so much for the feedback!
>
> 1. **Difference between truth-seeking and truthfulness/correctness**:
>    1. **Truthfulness/correctness on closed-ended questions**: Language models have been shown to hold *parametric beliefs* (beliefs stored in parameters) related to the question at hand [1]. When we measure model accuracy on common-sense logic or knowledge-based QA tasks, we are primarily testing the extent to which (i) the parametric beliefs match reality and (ii) the model honestly reports its parametric beliefs.
>    2. **Truth-seeking on open-ended questions**: Truth-seeking concerns the model’s ability to *form new, accurate beliefs at inference time through extended reasoning*, which is distinct from parametric beliefs. Frontier mathematics and scientific research questions, forecasting, academic peer review, and value reflection are examples of open-ended domains where this kind of extended reasoning is key. Since belief entrenchment is less likely to occur in mathematical reasoning, we use the other three, more subjective domains as our tasks.
> 2. **Clarifying the utility of Bayesian reasoning**: In Bayesian epistemology, evidence can be both external and internal. The capability of incorporating *external* evidence deals with search evidence to answer certain query; whilst the capability of incorporating *internal* evidence, Not all evidence is external to the reasoner; by coming up with novel arguments/counterarguments, logical deduction, etc, also helps LLMs (and humans too) to be better Bayesian reasoners. The language model can obtain evidence without external information, and therefore needs to perform Bayesian updating based on that evidence.
>    1. Example: Suppose that the language model is asked to forecast the match result between team A and team B in FIFA 2025, two teams that never played against each other before. The model may have memorized all past FIFA results in-parameter and already know the base win rates of A and B respectively, but by recalling the style of A and B and think about which style is advantageous in a head-to-head match, it will be able to update its belief, even though no external input is involved in the process.
>    2. Such discussion is rooted in Bayesian epistemology literature. Its precise term is “The Problem of Logical Omniscience”. It says that assuming a reasoner can only reassess their belief based on external evidence would equalize a reasoner to be “a logical omniscient”, meaning that if they hold a belief, they instantly know all the logical consequences of their belief or automatically believe all the theorems derived from their belief. Humans are obviously not like this, nor are LLMs [2].
> 3. **Whether training already ensures calibration**: Empirically, this is shown to not be the case [3]. Theoretically, this is likely because (i) models learn their belief updating strategies from human demonstrations in pre- and post-training datasets, while the humans themselves exhibit a wide range of cognitive biases including belief entrenchment [4]; and (ii) many post-training datasets train the model to do things entirely different from truth-seeking belief updating, e.g. unquestioningly follow user instructions, which has been shown to break the distribution-matching property of the cross-entropy loss in pretraining-stage autoregression [5].
> 4. **Clarifying what questions we aim at with the martingale metric**:
>    1. The forecasting questions that are resolved after the LLM knowledge cut-off date are examples of questions we think believe entrenchment hurts performance, similar to confirmation bias that hurts people’s judgment. While the model can’t answer those forecasting questions *in full* just with parametric belief, its parametric belief (e.g. “Team A has a historical win rate of 70% while team B only has 50%, so it seems like A has an advantage...”) serves as an informative starting point for reasoning (“...but B has a good shooter while A’s goalkeeper is subpar”). We are therefore interested in whether the parametric belief unduly dictates later belief updates, aka belief entrenchment.
> 5. **On Figure 2**: Thank you for the feedback. We are replacing it with more informative infographics about how exactly the Martingale metric is computed.
>
> ### Reference
>
> [1] Wentao Zhu et al., Language Models Represent Beliefs of Self and Others
>
> [2] Robert Stalnaker, The Problem of Logical Omniscience
>
> [3] Andrea Yaoyun Cui et al., Do Language Models Have Bayesian Brains? Distinguishing Stochastic and Deterministic Decision Patterns within Large Language Models
>
> [4] Aileen Oeberst et al., Toward Parsimony in Bias Research: A Proposed Common Framework of Belief-Consistent Information Processing for a Set of Biases
>
> [5] Robert Kirk et al., Understanding the Effects of RLHF on LLM Generalisation and Diversity

---

> > ### Comment · Reviewer_NJVu · 2025-08-05
> >
> > Thank you for your detailed response and for clarifying my concerns. I would suggest detailing the differences between truth-seeking and truthfulness clearly in the camera-ready manuscript to avoid further confusion.

---

> ### Author Response · Authors · 2025-08-09
> **The concern over LLM judge calibration**
>
> Here is our new results to address reviewer's early concern on LLM judge calibration. We would welcome reassessment if the reviewer finds the newly acquired results informative. Thank you!
>
>
> We would like to to start by noting that LLM calibration has been a notoriously difficult problem. Previous research has demonstrated that LLMs overall are poorly calibrated (in particular, they have strong tendency to overstate self-reported confidence; and token prob can go extremely high for binary problems, which does not represent true confidence) [1]. Findings reported in literature stays consistency with our experience of self-reported confidence and token probability.
>
>
> Instead, we move on to the **"LLM-as-judge" approach as a scalable alternative to avoid the problem of over-stated confidence and extreme token probability**. However, LLM-as-judge approach has its own biases (e.g., they may repeat human falsehood; or they may prefer their own answers) [2,3]. Hence, we conduct cross-judge consistency evaluation. Specifically, we construct pairs of judge (e.g., GPT-4o VS Gemini-2.5-pro, GPT-4o VS DeepSeek-v3) and see how much their belief evaluation correlates to each other. Note that we've acquired GPT-4o data for all batches but only a few batches for each of other judges (from 3 to 48 batches), so we set up GPT-4o as default judge for all comparison.
>
>
> Moreoever, **we use small batch of human-evaluation data to validate LLM judge** (i.e., human-llm consistency evaluation). Specifically, we construct pairs of human-LLM judges (e.g., human evaluator 1 VS GPT-4o). Note that the amount of human evaluation data is a lot smaller than that of LLM's, due to limited time. Nonetheless it's large enough (e.g., 195 and 173 belief samples) to give us a robust intuition of human-llm agreement on belief evaluation.
>
> Full results can be seen below:
>
> | Rank | Judge              | Batches | Problems | Belief Samples | Pearson r  | Spearman r | Interpretation     | p-value |
> | ---- | ------------------ | ------- | -------- | -------------- | ---------- | ---------- | ------------------ | ------- |
> | 1    | **Human-eval-1**   | 2       | 20       | 195            | **0.8822** | **0.8770** | **Positive Large** | < 0.001 |
> | 2    | **DeepSeek-v3**    | 48      | 3,834    | 24,921         | **0.7774** | **0.7620** | **Positive Large** | < 0.001 |
> | 3    | **GPT-4.1-mini**   | 3       | 283      | 2,015          | **0.7581** | **0.7490** | **Positive Large** | < 0.001 |
> | 4    | **Gemini-2.5-pro** | 4       | 373      | 1,688          | **0.7460** | **0.7230** | **Positive Large** | < 0.001 |
> | 5    | **Human-eval-2**   | 2       | 18       | 173            | **0.7152** | **0.6812** | **Positive Large** | < 0.001 |
>
> Overall, we have **Sample Size**: 31,992 total belief evaluations across 4,538 problems; **all judge show large positive correlation with GPT-4o**; all results are statistically significant (P<0.001). Among them, one of the human judges achieves highest correlation with GPT-4o, and all LLM judges achieve higher agreement with GPT-4o than the other human evaluator. As a comparison, NeurIPS 2021 experiments shows ~0.58 correlation between paper acceptance decisions made by two committees [4].
>
> Reference
> [1] Yudi Pawitan, Confidence in the Reasoning of Large Language Models
>
> [2] Stephanie Lin, TruthfulQA: Measuring How Models Mimic Human Falsehoods
>
> [3] Arjun Panickssery, LLM Evaluators Recognize and Favor Their Own Generations
>
> [4] Alina Beygelzimer, Has the Machine Learning Review Process Become More Arbitrary as the Field Has Grown? The NeurIPS 2021 Consistency Experiment

---

### Official Review · Reviewer_ZBd9 · 2025-06-30

**Clarity:** 3
**Significance:** 3
**Originality:** 3
**Rating:** 5
**Confidence:** 3

**Summary:**

This paper proposes the Martingale Score as a metric to evaluate belief entrenchment in LLM reasoning by measuring how well models update their beliefs in response to new evidence. The authors argue that effective reasoning requires Bayesian-like belief updating and use the Martingale Score to assess whether LLMs achieve better performance through genuine reasoning or by relying on pre-trained parametric knowledge. The method is evaluated across multiple domains and LLM architectures.

**Questions:**

Questions:
1. How exactly was the experimental setup implemented? The procedure for computing Martingale Scores in practice needs clarification.
2. Was evidence presented to LLMs with probability proportional to its truth value under the prior distribution, as required by equation 2? Or were the CoT trajectories assumed to be evidence samples from the prior?
3. How many CoT samples were used for each evaluation? Is this a Monte Carlo estimation or single-sample interpretation?
4. How do Martingale scores computed from judge assessments compare to those computed from model token probabilities?
5. Could you look at how Martingale scores correspond to different synthetic non-Bayesian update rules to provide intuition for various score ranges?
8. I don't understand how Figure 1 demonstrates belief entrenchment - could you clarify this illustration?

Comments:
1. The claim that "Bayesian reasoning is optimal" and that "effective reasoning requires Bayesian rationality" should include citations supporting these fundamental assumptions.
2. The statement about outcome-based evaluation being insufficient seems questionable - in expectation, with sufficient evaluation settings, shouldn't this be adequate since Bayesian reasoning is optimal?
3. While the concern about parametric recall limiting real-world utility is valid, there's a possibility that LLMs have sufficient parametric knowledge that belief entrenchment causes no real performance degradation.
4. Typo on line 171: extra space before comma.

**Ethical Concerns:**

["NO or VERY MINOR ethics concerns only"]

**Final Justification:**

The authors have addressed my main concerns with their response. I am satisfied by their proposed additions to the camera-ready version and I have adjusted my score accordingly.

**Limitations:**

The paper has a dedicated discussion on the method's limitations. It could additionally include:
1. The reliance on judge models for belief assessment and potential errors or biases this introduces.
2. The challenge of ensuring proper prior distributions in experimental setup
3. The scope of domains and question types where this metric is applicable. Currently evaluatied on Bernoulli distributions.

**Quality:**

3

**Strengths And Weaknesses:**

Strengths:
1. The paper addresses an important and novel question about the nature of LLM reasoning.
2. The Martingale Score provides a principled mathematical framework grounded in probability theory for measuring belief updating.
3. The authors test for statistical significance and report p-values < 0.05, demonstrating proper statistical methodology.
4. The evaluation spans multiple domains and LLM architectures, providing some generalizability.
5. The connection between Bayesian reasoning optimality and the proposed metric is theoretically motivated.
6. The paper evaluates multiple judge models to ensure robustness of belief assessment.

Weaknesses:
1. The experimental setup is not clearly described in the main body or even after reading Appendix B. Critical details are missing about how the Martingale Score was actually computed in practice.
2. There's a fundamental concern about the experimental procedure: The expectation in equation 2 must be with respect to the prior distribution, meaning evidence should be shown to the LLM with probability equal to that evidence being true under the prior. \delda b = 0 in expectation, how was this expectation approximated?
3. It's unclear if this was how the experiment was performed.
4. The use of judge models introduces serious a potential concern: The authors may be reporting systematic bias in judges' assessments of LLMs' beliefs rather than genuine belief entrenchment in LLM reasoning.
5. Key experimental details are missing: How many questions per domain? How many trajectories per question? How many reasoning steps per trajectory on average? How many CoT samples were used for Monte Carlo estimation (if this is indeed what's happening)?
5. The interpretation of chain-of-thought sampling is ambiguous. Is each CoT sample treated as sampling facts from the prior for Monte Carlo estimation of equation 2, or is a single CoT used with its final result interpreted as the expectation? I do not think the latter interpretation is correct.
6. The paper lacks comparison to token probability distributions over statements. Does belief assessment from judges differ from actual model probabilities? What are the Martingale scores when using model probabilities directly? The authors justify not using logits stating "LLMs are often poorly calibrated: their confidence scores—whether expressed through token probability or self-reported scores—does not reliably reflect their true degree of belief." What is defined to be their "true belief"? This needs justification or reference. Examining the token probability would remove biases introduced by judges.
7. I found the figures confusing more than helpful when trying to understand the experimental setup.

The paper presents an interesting theoretical contribution with the Martingale Score and addresses an important question about the nature of LLM reasoning. However, the experimental setup is insufficiently clear, making it difficult to properly evaluate the core claims. I would be happy to raise my score if my concerns are addressed.

---

> ### Author Rebuttal · Authors · 2025-07-31
>
> Thanks reviewer for raising constructive comments, pointing out weakness and limitations, and asking thought-provoking questions. The following is our attempt to address those comments and questions.
>
> ### Experimental setups and processes and the computation of martingale score (Weakness 1, 2, 3, 5, Q3)
>
> Thanks for pointing out all the weaknesses and asking questions about experimental setups.
>
> - The unclear description of experimental setup and missing out of martingale score calculation in practice:
>   - As mentioned in section 4.1, we use Ordinary Least Squares (OLS) method to approximate martingale score in practice. Still, the detail here may still appear inadequate to reviewers and broader audiences. We suspect it might be because we did not link the definition of martingale score to the experiment setup. In our experiments, we calculate the martingale score per setup, where we include 99 questions (n=99; e.g., forecasting questions), b_prior is calculated when reasoning starts, which reflects our best attempt to capture a parametric belief of LLMs; b_posterior is calculated after inference or certain amount of reasoning steps; and b_update is the difference between both.
> - The experimental procedure:
>   - Key experimental details requested: There are around 200 questions per domain, 1 trajectory per question, and ~ 5 reasoning steps per trajectory. We use 200 different problems per setup for each Monte Carlo Estimation. In other words, calculate expectation of model belief across questions. Thanks for pointing out the missing details here, and they will be added into the final manuscript.
>
> ### The concerns over judge models and token probability (W4, Q4)
>
> We thank the reviewer for this insightful comment. Our use of an external judge model is a deliberate methodological choice to address the well-known challenges of reliably eliciting beliefs from LLMs. Our approach is validated by new, large-scale tests showing high consistency across different judge models.
>
> Of course. Here is a revised rebuttal that is more concise, incorporates your supervisor's feedback, and addresses the status of the cross-judge validation.
>
> We thank the reviewer for this insightful comment. Our use of an external judge model is a deliberate methodological choice, made because direct belief elicitation from LLMs is notoriously unreliable. Our approach focuses on measuring expressed belief, and while ensuring cross-judge consistency is an important validation step, the core method remains a pragmatic solution to the challenges of probing a model's "true" belief state.
>
> The Unreliability of Direct Belief Elicitation
>
> We initially attempted to measure belief using direct methods like token probabilities. This approach was unworkable because LLMs consistently produce extreme and poorly calibrated probabilities for binary questions (e.g., 99.9%) [1]. This is a recognized challenge in the field of ML calibration. These extreme values do not reflect a model's true epistemic state; in our experiments, models expressing near-certainty would often reverse their positions easily. This makes token probability an unsuitable tool for measuring nuanced belief updates.
>
> The LLM-as-Judge Approach
>
> To overcome this, we use an external judge model. This shifts the focus from an unreliable "internal belief" to the model's expressed belief as interpreted by an independent observer. The primary concern here, which the reviewer rightly notes, is the reliability of the judge itself. We agree that a comprehensive, cross-judge reliability analysis is an important step to fully validate this evaluation paradigm.
>
> However, the core logic of using an external judge is sound. We use a courtroom analogy: the reasoning LLM is the "suspect" asked to think out loud, and the judge model is the independent observer assessing that monologue for bias. This allows us to measure entrenchment in the expressed reasoning—which is most relevant for human-AI interaction—while sidestepping the thorny question of "what does an LLM truly believe?"
>
> This focus is suited for studying how models update on internal evidence—the arguments and logical deductions they generate themselves. This allows us to explore a key aspect of their behavior related to the "Problem of Logical Omniscience" in Bayesian epistemology, justifying the use of an external judge to assess the reasoning process.
>
> ### The question of whether sufficient parametric knowledge recall is sufficient for practical use (comment 3).
>
> - While the intuition that a model with sufficient parametric knowledge makes it a good Q&A assistant, such a model may be inadequate to the needs of the real-world, as the knowledge production in real-world is a dynamic process whereas the parametric knowledge, however good it might be, is static. When deployed in the real-world, such a model faces practical limitations. LLMs deployed in the real-world are expected to give user evidence-based answers (evidence acquired from search or user-interactions), rather than only share prior-conforming ones. In contrast, LLMs that suffer from belief entrenchment may experience performance degradation because of prior-conforming behaviors (as shown in forecasting).
>
> ### The scope of domain where the metric is applicable (limitation 3)
>
> - While it is tested on Bernoulli distribution, martingale metric is intended to be an unsupervised, domain/distribution-agnostic metric. It is tested in forecasting questions and peer review because in these two problem domains there are ground truth labels to validate martingale scores. Similar to confirmation bias that harms human judgment which is not domain-specific, belief entrenchment hurt LLM performance, too, as we demonstrated in forecasting, As research progresses, we hope to find ways to validate martingale score in subjective domains, or domains where ground truth labels simply do not exist (Two domains we plan to run: OpenReview but to assess the relation between martingale score and citation number (as this more closely aligns with collective voted judgments) and Reddict r/AITA (also collective voted judgments).
>
> ### The question of whether evidence LLMs see is already biased (Q2)
>
> - As mentioned above, in Bayesian epistemology evidence can be either internal or external, and for practical reasons we primarily focus on internal evidence in our current experimental setups. To directly answer the reviewer's question: the internal evidence (the reasoning steps) are sampled from the prior and its context, and if it is biased, then this is evidential that belief entrenchment exists: model may always output prior-conforming reasonings to answer user questions.
>
> ### Conceptual discussion on Bayesian rationality (comment 1)
>
> - The request for “more literature to support the optimality of Bayesian rationality” is fair. We thank the reviewer for pointing this out. Bayesian reasoning is regarded as optimal because it provides a quantitative framework to incorporate both prior and new evidence in forming beliefs and it is able to handle uncertainty [2]. Conceptual arguments (e.g., “Dutch Book Argument” [3]: The claim that "Bayesian reasoning is optimal" and that "effective reasoning requires Bayesian rationality") and ample empirical evidence support the effectiveness of Bayesian reasoning (e.g., financial modeling, jury decision,medical diagnosis) [4] We thank reviewer for this valuable comment.
>
> ### Assigning intuitive labels to Martingale Score (Q5)
>
> We thank the reviewer for pointing out those important omissions, or otherwise lacking clarity of experimental setup in our paper submission and confusion with figure 1. And thanks for pointing out typos in the manuscript. All those overlooked details will be added into the final publication.
>
> ### References:
>
> [1] Yudi Pawitan & Chris Holmes, Confidence in the Reasoning of Large Language Models
>
> [2] C Howson & P Urbach, Scientific reasoning: the Bayesian approach
>
> [3] Dutch Book Arguments, Stanford Encyclopedia of Philosophy
>
> [4] Aaron M. Ellison, Bayesian inference in ecology

---

> > ### Comment · Reviewer_ZBd9 · 2025-08-07
> > **Re: Rebuttal by Authors**
> >
> > The authors have addressed my main concerns with their response. I am satisfied by their proposed additions to the camera-ready version and I have adjusted my score accordingly.

---

> ### Author Response · Authors · 2025-08-09
> **Addressing the concern that using LLM judge may introduce judge bias**
>
> Thanks for your earlier comment. In the following official comment we would like to share our judge-consistency and human-evaluation validation to address the concern over judge bias. We would welcome reassessment if the reviewer finds the newly acquired results informative.
>
> We explained our rational of using LLM as judge models to overcome the problems with self-report and token probability [1]. As the reviewer rightly pointed out, the LLM-as-judge approach may introduce judge bias in the process of evaluating belief entrenchment. Here we include a note on the conceptual side of belief evaluation and our empirical results of testing different judges.
>
> On the conceptual side, We treat (assume) the judge model as a reliable & independent observer of the reasoner's “monologue” and assessing whether the reasoning process suffers from belief entrenchment. Using a court analogue here: with this method, the judge model,  just like a "court clerk", simply asks the “suspect” to "think out loud" about whether they are guilty. If the suspect's internal monologue (the CoT) is a biased stream of "evidence" aimed at proving their innocence, a judge who answer how the binary questions are resolved faithfully based on the records of monologues, will be one-sided. By doing so, we can evaluate whether model + LLM judge as a multi-agent systems, suffer from belief entrenchment. And of course, here it comes to the question of judge bias. If belief entrenchment is a judge bias rather than model's, then changing judge models would remove such artifacts. However, in our experiments, judge evaluations are strongly correlated.
>
> In our cross-judge consistency and human-LLM agreement anslysis, all judges (LLMs or humans) show large strong correlation with GPT-4o.
>
> **Cross-judge consistency evaluation**: we construct pairs of judge (e.g., GPT-4o VS Gemini-2.5-pro, GPT-4o VS DeepSeek-v3) and see how much their belief evaluation correlates to each other. Note that we've acquired GPT-4o data for all batches but only a few batches for each of other judges (from 3 to 48 batches), so we set up GPT-4o as default judge for all comparison.
>
> **Human-LLM agreement evaluation**: we use small batch of human-evaluation data to validate LLM judge (i.e., human-llm consistency evaluation). Specifically, we construct pairs of human-LLM judges (e.g., human evaluator 1 VS GPT-4o). Note that the amount of human evaluation data is a lot smaller than that of LLM's, due to limited time. Nonetheless it's large enough to give us a robust intuition of human-llm agreement on belief evaluation.
>
> Full results can be seen below:
>
> | Rank | Judge              | Batches | Problems | Belief Samples | Pearson r  | Spearman r | Interpretation     | p-value |
> | ---- | ------------------ | ------- | -------- | -------------- | ---------- | ---------- | ------------------ | ------- |
> | 1    | **Human-eval-1**   | 2       | 20       | 195            | **0.8822** | **0.8770** | **Positive Large** | < 0.001 |
> | 2    | **DeepSeek-v3**    | 48      | 3,834    | 24,921         | **0.7774** | **0.7620** | **Positive Large** | < 0.001 |
> | 3    | **GPT-4.1-mini**   | 3       | 283      | 2,015          | **0.7581** | **0.7490** | **Positive Large** | < 0.001 |
> | 4    | **Gemini-2.5-pro** | 4       | 373      | 1,688          | **0.7460** | **0.7230** | **Positive Large** | < 0.001 |
> | 5    | **Human-eval-2**   | 2       | 18       | 173            | **0.7152** | **0.6812** | **Positive Large** | < 0.001 |
>
> Overall, we have **Sample Size**: 31,992 total belief evaluations across 4,538 problems; **all judge show large positive correlation with GPT-4o**; all results are statistically significant (P<0.001). Among them, one of the human judges achieves highest correlation with GPT-4o, and all LLM judges achieve higher agreement with GPT-4o than the other human evaluator. As a comparison, NeurIPS 2021 experiments shows ~0.58 correlation between paper acceptance decisions made by two committees [4].
>
> Reference
>
> [1] Yudi Pawitan, Confidence in the Reasoning of Large Language Models
>
> [2] Alina Beygelzimer, Has the Machine Learning Review Process Become More Arbitrary as the Field Has Grown? The NeurIPS 2021 Consistency Experiment

---

### Official Review · Reviewer_ofLg · 2025-07-01

**Clarity:** 4
**Significance:** 3
**Originality:** 4
**Rating:** 5
**Confidence:** 4

**Summary:**

### Problem
- This paper investigates an issue in LLM reasoning: **belief entrenchment**. Intuitively, this refers to the phenomenon where a language model fails to revise its beliefs based on new evidence, instead clinging to its original stance (prior belief). For example, when asked “Will AI destroy humanity?”, a model might initially lean toward “No.” Even after being presented with multiple counterexamples, it still concludes “No,” indicating that its belief update was not driven by evidence—this is belief entrenchment. Ideally, a reasoning process should be evidence-based; conclusions should depend on facts, not on the model’s initial belief. The central question the authors pose is whether LLM reasoning genuinely reflects **belief updates based on new evidence**, or if it **can be predicted by prior beliefs**.

### Framework
To evaluate this, the paper introduces a concept from Bayesian statistics—the **Martingale property**, which states that for a rational Bayesian agent, the **expected posterior belief should equal the prior belief**. In other words, the **change in belief (Δb)** should be **statistically unpredictable from the prior belief ($b_0$)**. If the update $Δb$ is systematically correlated with $b_0$ (e.g., the higher the prior, the larger the update), this suggests the model is reinforcing its prior, not responding to evidence.

To quantify this, the authors propose a new metric: the **Martingale Score**, computed via a simple **ordinary least squares (OLS)** regression:
$$Δb = β_1·b_0 + β_0 + ε$$
A non-zero $β_1$ indicates a deviation from the Martingale property—hence, evidence of entrenchment. OLS is chosen for its simplicity and robustness: it makes minimal assumptions about the data and provides a direct measure of predictability.

Rather than relying on internal token probabilities, which are quite unreliable, the paper uses a judge model (e.g., GPT-4o) to assign explicit belief scores $b ∈ \[0,1]$ to each model output. The belief update is obtained by giving the model some new information and using the judge model to evaluate the new belief score.

### Experiments
The authors conduct empirical evaluations across three real-world tasks:

1. **Forecasting future events** (Metaculus, Polymarket)
2. **Peer review decision-making** (ICLR OpenReview)
3. **Value-laden debates** (r/ChangeMyView)

They find that across most models and tasks, the **Martingale Score is positive**, indicating widespread belief entrenchment. Moreover, in domains where ground truth is available, the Martingale Score has slight positive correlation with the **Brier Score** (a measure of prediction error), suggesting that entrenchment not only exists but also is likely to **degrade reasoning accuracy**.

**Questions:**

1. How robust is the belief extraction via judge models? For me, the examples in Figure 1 and Figure 3 are quite difficult to extract their corresponding belief. Are belief scores consistent across different judges?
2. What is the scale of the datasets used? How many samples are there per setup? Given that the Martingale Score relies on regression estimates, is there a lower bound on the number of samples required for the score to be statistically meaningful?
3. What exactly does the t-test in Table 1 evaluate? If it is testing whether β₁ ≠ 0, then in many forecasting and ChangeMyView settings the p-values are > 0.05. Should we then interpret those Martingale Scores as statistically insignificant—possibly consistent with no entrenchment?
4. The interpretation of positive vs. negative Martingale Scores is somewhat unclear. The paper focuses primarily on β₁ > 0 (entrenchment), but what about cases where β₁ < 0? What does a negative Martingale Score imply in terms of belief updating behavior?
5. Figure 5 is hard to interpret. Could you provide a more intuitive explanation of what the regression is doing and how to read the plotted coefficients across model specifications?

**Ethical Concerns:**

["NO or VERY MINOR ethics concerns only"]

**Final Justification:**

I have decided to remain my score for the novelty of the proposed question, the adequate use of the Martingale property as a theoretical lens and the solid experiment setup.

**Limitations:**

The paper evaluates whether belief entrenchment exists in LLMs, but it does not explore why LLMs exhibit this behavior or what mechanisms drive it.

**Quality:**

4

**Strengths And Weaknesses:**

### Strength
1. [Significance] The paper addresses an underexplored yet important question: whether LLMs update beliefs based on evidence or merely reinforce prior assumptions.
2. [Originality] The use of the Martingale property as a theoretical lens is elegant and novel. The proposed Martingale Score is a well-motivated and principled metric that operationalizes belief entrenchment in a Bayesian framework.
3. [Clarity] The paper is well-written, with a clear structure and smooth narrative. Definitions, assumptions, and experimental design are all presented with clarity.
4. [Quality] The experimental setup is comprehensive. The three tasks—forecasting, peer review, and value-based reasoning—seem quite diverse and comprehensive reasoning scenarios for me.

### Weakness
1. Some empirical results are not fully convincing. For instance, the positive correlation claim in Figure 4 appears weak and potentially overstated.

---

> ### Author Rebuttal · Authors · 2025-07-31
>
> Thank you so much for the thoughtful feedback!
>
> 1. **Experimental setup details**: There are around 200 questions per domain, 1 trajectory per question, and 2-20 reasoning steps per trajectory. We use 200 different problems per setup for each Monte Carlo Estimation. In table 1, it can be seen that almost all CoT entries have statistically significant Martingale scores. For Debate, while only some of the individual entries are statistically significant, on aggregate, we are able to show it significantly reduces belief entrenchment compared to CoT (Figure 7a).
> 2. **The interpretation of t-test results in table 1**: The t-tests attempt to reject the null hypothesis that **no belief entrenchment exists** (i.e. the Martingale score is zero at the limit of infinite samples). When p>0.05, it should be interpreted as that no credible evidence for belief entrenchment exists.
> 3. **The interpretation of positive and negative Martingale Scores**: When the Martingale score is negative **with p<0.05**, it should be interpreted as that the model tends to be overly “self-critical” - it tends to update *against* its prior belief - the opposite of belief entrenchment. This is observed in only 2 out of 108 setups, and **is very rare**. As a result, we believe it is of limited value to study these rare cases in depth.
> 4. **An intuitive explanation of figure 5**: The regression in Fig 5 answers the question “Does a higher (absolute value of) Martingale score causally worsens accuracy?”, where every sample is the (Martingale score, accuracy) pair of a cell in Table 1. To tell causality from mere correlation, we add *control variables* to the regression and thereby remove confounders.
>    1. Taken together, we have these variables in our regression:
>       1. **Explanatory variable**: abs(Martingale)
>       2. **Explained variable**: accuracy, i.e. Brier score
>       3. **Control variables**: Domain (forecasting/openreview/CMV), Reasoning Mode (CoT/debate), Prompt (No/CT/PC).
>    2. In Fig 5, from left to right, we gradually add more control variables. We watch for (i) **the increase in R2** (lower subplot), indicating how much of the accuracy variance has been captured by our control variables; (ii) **the decrease in p-value** (black text labels in upper subplot) of the regression coefficient on abs(Martingale); and (iii) **the sign of the regression coefficient** on abs(Martingale).
>       1. It can be seen that, when we have at least Domain + RM as controls, we have highly non-trivial R2 (0.27-0.4), p-values < 0.05, and the coefficient on abs(Martingale) is consistently positive.
>       2. This suggests that, **after adjusting for confounders (domain & reasoning mode), larger absolute value of the Martingale score (aka more deviation from the Martingale property) causally harms accuracy.**
> 5. Re “the positive correlation claim in Figure 4 appears weak and potentially overstated”:
>    1. For CoT on Forecasting:
>       1. **Robustness**: rho = 0.63; this is a consistent correlation
>       2. **Effect size**: A 0.1 increase in the Martingale score corresponds to approximately a 0.15 increase in Brier score, which is a huge drop in accuracy, **larger than the difference (0.101) between human crowd wisdom and dice-rolling**. [1]
>    2. For CoT on OpenReview:
>       1. **Robustness**: rho = 0.20, and after removing outlier rho=0.53; while this is smaller than in forecasting, this is likely due to the extreme noise in peer review labels [2].
>       2. **Effect size**: Similarly large compared to in forecasting.
>    3. For Debate: It’s worth noting that for most of the Debate entries in Table 1, their Martingale scores are not statistically significant. As such, we have no evidence suggesting belief entrenchment exists in such cases. This would be consistent with the observation that the regression lines are only weakly upward-sloping, as they are dominated by noise.
>
> ### Reference
>
> [1] Danny Halawi et al., Approaching Human-Level Forecasting with Language Models
>
> [2] Alina Beygelzimer et al., The NeurIPS 2021 Consistency Experiment

---

> > ### Comment · Reviewer_ofLg · 2025-08-07
> > **Thank you for your reply!**
> >
> > Thank you for your response, which clarified my concerns.

---

### Official Review · Reviewer_G3H9 · 2025-07-03

**Clarity:** 4
**Significance:** 4
**Originality:** 4
**Rating:** 5
**Confidence:** 4

**Summary:**

This paper introduces a novel, unsupervised metric called the "Martingale Score" to quantify belief entrenchment, which is a model's tendency to adhere to its prior beliefs rather than rationally updating them with new evidence. The score is derived from the Martingale property in Bayesian statistics, which posits that a rational agent's belief updates should be unpredictable from their current belief. The authors operationalize this by measuring the linear relationship between a model's prior belief and its subsequent belief update. Through extensive experiments across various LLMs, reasoning techniques (CoT, Debate), and domains (forecasting, value-laden questions, paper review), the paper demonstrates that belief entrenchment is a widespread phenomenon. Crucially, the authors validate the Martingale Score by showing it negatively correlates with ground-truth accuracy, confirming its utility as a proxy for the quality and truth-seeking ability of an LLM's reasoning process.

**Questions:**

The belief extraction process using a "judge" model is a critical dependency. Could you provide more detail on its robustness? For example, how much did the Martingale Scores vary when different families of models (e.g., Claude, Gemini) were used as the judge, and did these variations affect the paper's main conclusions?

The paper finds that the 'Debate' reasoning technique mitigates entrenchment. Does the Martingale Score provide any insight into why this happens? For instance, does debate lead to belief updates that are fundamentally less predictable from the initial stance, or does it simply moderate the initial beliefs, leading to smaller but still predictable updates?

The choice of a linear regression to define the Martingale Score is simple and interpretable. Did you explore potential non-linear relationships between prior belief and belief updates? Could a more complex model capture nuances of belief entrenchment that the linear coefficient might miss, or did the linear model prove to be the most effective empirically?

**Ethical Concerns:**

["NO or VERY MINOR ethics concerns only"]

**Final Justification:**

I read the author rebuttal and the discussion. The authors clarified the scope of the Martingale Score, explained why debate reduces entrenchment mainly by moderating update magnitude, and committed to add details on judge robustness, sample sizes, and effect size reporting, plus figure and caption improvements. These commitments address my questions about practical robustness and interpretation. I maintain my rating of Accept with confidence 4.

**Limitations:**

Yes

**Paper Formatting Concerns:**

None.

**Quality:**

3

**Strengths And Weaknesses:**

Strengths:

1. Originality and Significance: The paper tackles the critical and timely problem of confirmation bias in LLMs. The proposed Martingale Score is a highly original contribution, providing a principled, unsupervised, and domain-agnostic method to measure belief entrenchment. Grounding the metric in the Martingale property from Bayesian statistics gives it a strong theoretical foundation that is both elegant and powerful.

2. Quality and Rigor: The experimental design is thorough and robust. By testing across a wide range of models, domains, and reasoning paradigms, the authors provide compelling evidence for their claims. The validation of the Martingale Score against ground-truth accuracy (Brier score) is a key strength, demonstrating that the proposed metric is not just a theoretical construct but a practical indicator of reasoning quality.

3. Clarity: The paper is exceptionally well-written and easy to follow. Complex statistical concepts are explained intuitively, and the methodology is laid out clearly. The figures, particularly the conceptual illustration of the Martingale Score (Fig. 2) and the results showing its correlation with accuracy loss (Fig. 4), are highly effective at conveying the core message.

Weaknesses:

1. Belief Elicitation Method: The framework's reliance on a separate "judge" LLM to extract a scalar belief score from a model's natural language reasoning is a potential weakness. This introduces a dependency on the judge model's own characteristics and could be a source of noise or bias. While the authors state they checked for consistency, the paper would benefit from a more detailed analysis of the robustness of the Martingale Score to the choice of judge model.

2. Focus on Diagnosis over Mitigation: The paper's primary contribution is diagnostic—identifying and measuring belief entrenchment. While it finds that certain existing techniques (e.g., debate) can mitigate entrenchment, it does not propose novel interventions based on the Martingale Score itself. The authors acknowledge this in their future work, but the current work stops short of providing new solutions to the problem it identifies.

---

> ### Author Rebuttal · Authors · 2025-07-31
>
> Thank you for the thoughts. We really appreciate your feedback!
>
>
>
> 1. **Focus on diagnosis over mitigation:** While our experiments cover only diagnosis, we believe our contribution extends to mitigation too. A key difficulty for comprehensively correcting model cognitive biases is that *doing so requires ground-truth information,* which is unavailable in almost all domains of interest (domains involving subjective judgments). By proposing the Martingale score, a fully unsupervised & domain-agnostic metric for belief entrenchment, we demonstrate both its promise as an evaluation metric, and, as a corollary, its potential as a training signal.
> 2. **Nonlinearities**: We have experimented with ways to handle outliers in our regression, which is a primary cause of nonlinearities in data. We also tried quadratic regression. However, non-linear regressions in general lack a *single* coefficient or value that reliably points to “causal contribution of prior to belief updates”; this makes the design of the Martingale score difficult. We also have a relatively small sample size, which risks overfitting when using parameter-rich models of regression.
> 3. **Does the Martingale Score provide any insight into why debate helps?**
>    1. We think your hypothesis turns out correct! Analysis of 78 reasoning trajectories shows that instead of making AI reasoning fundamentally better, debate mostly just makes belief updates more conservative. **This shows that** **removing** **belief entrenchment is still a hard open problem, and we are glad that we unveiled this problem with our evaluations.**
>    2. Specifically: When we looked at bias scores regularized by posterior variance vs belief change variance (Var[delta]), both give similar model fits (R² around 0.19-0.27). But the Var[delta] models reveal something interesting - debate's main job is keeping belief changes from going wild rather than actually improving calibration.
>       - Here, the motivation for regularizing with Var[delta] is that, we want to regularize the amount of reasoning/updating that gets down, to rule out the possibility that debate merely reduces updating.
>    3. The correlation patterns support this conclusion. In forecasting tasks, chain-of-thought reasoning shows a strong bias-accuracy relationship (ρ=0.63, p=0.002); basically, when it's biased, performance tanks. But, for self-debate, it's nearly flat correlation (ρ=0.06, p=0.80). This suggests that debate doesn't actually eliminate the bias; instead, it's the bias measurements being overwhelmed by stochastic noise, likely due to insufficient sample size as a result of our compute budget.

---

> > ### Comment · Reviewer_G3H9 · 2025-08-09
> >
> > I read the author rebuttal and the discussion. The authors clarified the scope of the Martingale Score, explained why debate reduces entrenchment mainly by moderating update magnitude, and committed to add details on judge robustness, sample sizes, and effect size reporting, plus figure and caption improvements. These commitments address my questions about practical robustness and interpretation. I maintain my rating of Accept with confidence 4.

---

### Note · Authors · 2025-08-13

We thank reviewers, ACs, and SACs for engaging with this paper! The Martingale Score is an unsupervised metric to evaluate belief entrenchment - how much LLM disproportionately reinforces its prior opinion during reasoning, instead of conducting evidence-based Bayesian updating.

We thank reviewers for recognizing the originality of martingale score, the importance of the belief entrenchment problem, the rigor of experiments, and the theoretical connection to Bayesian reasoning optimality. The validation of martingale score against ground truth also makes Martingale Score practically relevant. Below we summarize our responses to the major concerns over *robustness* and *generality* of Martingale Score.

## Robustness

There are concerns over using judge models to evaluate beliefs, namely on judge bias and calibration. In the later stage of rebuttal, we perform cross-judge consistency evaluation and use human evaluation to validate the LLM judge. We found that all judges show large positive correlation ($\rho\geq 0.75$) with GPT-4o (default), and both human annotators are also strongly and positively correlated with GPT-4o’s ($\rho=0.88, 0.72$). All results are statistically significant. We hope reviewers and ACs could consider those promising results when giving final scores.

## Generality

Martingale Score, grounded in theory, is intended to be a generally applicable metric to evaluate belief entrenchment in LLM reasoning (corresponding to confirmation bias in humans). The validation of martingale scores is hard because we need domains where ground truth is inaccessible by LLMs but accessible by human evaluators. This leaves us only forecasting datasets (and to a lesser extent, OpenReview). Fortunately, LLM-based forecasting is a general methodology applied in many *problem domains*, and such capability - weighing in evidence in light of uncertainties, is central to general reasoning. Thus, effectiveness on Forecasting is a good indicator of the Martingale score’s effectiveness on reasoning.

Besides, our extended experiments did show more promising results with OpenReview. In a follow-up study, we are conducting Martingale training to see its performance comparison against training with ground-truth labels. Promising results in the follow-up study would further address this concern.

We thank reviewers for raising concerns over the robustness and generality of Martingale Score. We would welcome re-assessment if these results address major concerns.

---

### Decision · Program_Chairs · 2025-09-17

**Decision:**

Accept (poster)

**Comment:**

The paper investigates the "belief entrenchment phenomenon" in Large Language Models (LLMs). This phenomenon refers to a model's tendency to resist updating its beliefs in light of new evidence, instead sticking to its prior beliefs. To assess this phenomenon, the authors introduce an unsupervised, regression-based metric called the Martingale Score. This score quantifies deviations from the Bayesian Martingale property, which posits that belief updates should be unpredictable based on previous beliefs. Instances of such deviations indicate irrational entrenchment. The authors discover that these violations are prevalent across various LLM models, reasoning techniques (including Chain-of-Thought and Debate), and problem domains. Furthermore, they validate the Martingale Score through experiments that demonstrate its ability to predict ground-truth accuracy in domains where valid labels are available.

In total, this paper received five reviews, with four recommending acceptance and one recommending rejection. All reviewers concur on the significance of the study's motivation, the originality of the Martingale Score, and the quality of the paper. However, the statistical significance of belief entrenchment and the scoring metrics have been contentious topics during the rebuttal phase.

In summary, I recommend acceptance of the paper, but I strongly encourage the authors to consider the feedback provided by Reviewer #K3qf. Additionally, improvements could be made in the clarity of the introduction and the readability of the figures.